# A Critical Step to Using a Parametric Array Loudspeaker in Mobile Devices

**DOI:** 10.3390/s19204449

**Published:** 2019-10-14

**Authors:** Hongmin Ahn, Kyounghun Been, In-Dong Kim, Chong Hyun Lee, Wonkyu Moon

**Affiliations:** 1Department of Mechanical Engineering, POSTECH, Pohang 37673, Korea; idealcircuit@postech.ac.kr (H.A.); khbeen@postech.ac.kr (K.B.); 2Department of Electrical Engineering, Pukyoung National University, Busan 48513, Korea; idkim@pknu.ac.kr; 3Department of Ocean System Engineering, Jeju National University, Jeju 63243, Korea; chonglee@jejunu.ac.kr

**Keywords:** parametric array, directional speaker, piezoelectric micro-machined ultrasonic transducer

## Abstract

A parametric array (PA) loudspeaker is a highly directional audio source that might grant one's convenience if it is used with mobile devices. However, conventional PA loudspeakers is almost impossible to apply in mobile devices using a battery because of the large power consumption and large device size. In this study, a PA loudspeaker system (PALS) was fabricated and evaluated to show that those difficulties could be overcome to apply it to mobile devices. In order to construct a PALS for demonstration, a power amplifier and signal-processing unit should also be properly designed and built. The PA source transducer should also be designed and built for a mobile device application. These components were integrated into a single PALS. The PALS generated a 125-dB primary wave and 62 dB of a different frequency wave (DFW) through the PA at 0.45 m in a 3 m × 3 m × 2 m semi-anechoic chamber. We confirmed that the half-power bandwidth (HPBW) formed a 6° beam at 83 kHz of DFW and 90 kHz of the primary wave (PW), and the HPBW formed a 7.3° beam at 5 kHz of DFW and a 7.1° beam at 10 kHz of DFW, respectively. Lastly, the power required was 6.65 W without a matching circuit, and 3.25 W with such a circuit.

## 1. Introduction

The parametric array (PA) loudspeaker is a highly directional audio source that may be useful in special environments such as museums, exhibitions, and galleries, as well as for multi-language teleconferencing not widely known to the public. Since this loudspeaker generates audible sounds through the PA phenomenon via finite amplitude acoustic progressive waves [1,2], the directivity of the radiating audible sounds is much higher than those from conventional loudspeakers of a comparable size. However, because the PA phenomenon is a second-order nonlinear effect observed during propagation of progressive acoustic beams with finite amplitudes, the generated audible sounds are very weak and the waveforms are difficult to control. 

Yoneyama et al. first proposed the PA source as a loudspeaker in 1983 [1], after Bennett and Blackstock reported their observation of in-air PA sounds in 1975 [3]. In 1998, 15 years after the original report of Yoneyama et al., the PA loudspeaker was commercialized and there are now several companies who manufacture and sell PA loudspeakers to the public [4]. As discussed, the PA loudspeaker indirectly generates audible sounds through a nonlinear interaction in finite amplitude acoustic progressive waves. Hence, its power efficiency is extremely low and its indirectly-generated target audible sounds are difficult to control via input signals. As a consequence, typical PA loudspeakers on the market usually have high power consumption, low sound quality, and a large size relative to their maximum output sound level. However, the extremely high directivity of the output audible sounds makes PA loudspeakers a highly specialized audio source.

A PA loudspeaker can be realized by combining several technologies: a power amplifier for wideband ultrasonic signals, signal processing technology to create the ultrasonic signals required for generation of the target audible sounds, and a sound radiator for highly directive ultrasonic sounds with a large amplitude and wide frequency bandwidth [3]. Many studies on signal processing techniques have been reported for improving the quality of audible sounds from the PA loudspeakers because its sound quality does not seem as good as the conventional loudspeakers [1,5]. However, it might be another good way to improve sound quality that a more adequate PA source transducer would be developed for PA loudspeaker applications [6,7]. In conclusion, the PA loudspeaker cannot yet be considered a highly successful audio source with a wide range of applications.

This study investigates the possibility of applying the PA loudspeaker to mobile devices. Audible sounds with high directivity may be useful when a mobile device plays multimedia content in public places. However, the large power consumption of conventional PA loudspeakers (usually more than 60 Watts) makes it extremely difficult to operate it within a mobile device by using a battery as a power source [4,7,8,9]. The applicability of the PA loudspeaker to mobile devices is, therefore, highly dependent on the extent to which its power consumption can be reduced. While the signal processing unit might consume only a small amount of power due to the well-developed digital microprocessors and peripheral chips, the power consumption of conventional power amplifier units usually depends on the output power it should generate. Hence, its power efficiency varies with the operating conditions [10]. However, the power consumption of the ultrasonic source transducer normally constitutes a large proportion of the total power consumption of a PA loudspeaker system, since the power efficiency is very low, at usually less than 10%. Therefore, it is important to reduce the power consumption of the source transducer by increasing its power efficiency.

In 2013, Je et al. reported that an ultrasonic uni-morph-type radiator composed of a thin plate and a PZT layer, which is a type of piezoelectric micromachined ultrasonic transducer (pMUT), could radiate finite amplitude ultrasonic sounds efficiently at up to 80% power efficiency [10]. In the same paper, they also proposed a technique for constructing an array of pMUTs for generating ultrasonic sound beams with a wide frequency bandwidth and higher power efficiency. They demonstrated that the fabricated array of pMUTs could generate PA audible sounds at low power consumption [10]. Hence, array pMUTs should be adopted as a component of the target PA loudspeaker for mobile devices such as notebook computers or, if the power consumption is low enough, smart phones. 

To investigate the possibility of application of a PA loudspeaker to mobile devices, a PA loudspeaker system should be built and evaluated to establish whether its total power consumption is sufficiently low. To build a PA loudspeaker system for this purpose, three components should be designed and fabricated appropriately: the signal processing unit, the power amplifier for wideband ultrasonic signals, and the PA source transducer with adequate radiation characteristics, which is the pMUT array in this paper. After building the PA system for testing, the total power consumption should be checked while the built system should generate PA sounds at an appropriate volume. The feasibility of the system might be proven if the total power consumption of the three components is sufficiently low so that a conventional battery in a mobile device can provide the power. The cost of the PA loudspeaker is influenced by the design and is built of the three components. This may constitute a limitation on the overall design. In this study, the power amplifier and signal processing unit were designed so that they could be built at a moderate cost, while the PA source transducer is designed and fabricated based on the design and fabrication methods for the pMUT array radiator developed by the research group [10].

## 2. Requirements of the Parametric Array Loudspeaker

A PA loudspeaker system is normally composed of three components, as illustrated in Figure 1. The components include an ultrasonic source transducer unit, a signal-processing unit, and a power amplifier unit. The signal-processing unit produces the ultrasonic sound signals directly radiating from the ultrasonic source transducer unit, which can be obtained by transforming the target audio signals through an appropriate kernel, which is similar to amplitude modulation (AM). The power amplifier unit drives the source transducer according to the input ultrasonic signals after amplifying their electric power. The finite amplitude ultrasonic sounds radiated from the source transducer, which would then generate audible sounds, frequently known as PA sounds, through nonlinear acoustic interaction effects known as the natural demodulation process [11].

The sound pressure level (SPL) of the PA sounds, which may be viewed as a by-product of the directly radiated primary ultrasonic sounds, is usually 40 dB less than that of the ultrasonic sounds directly radiating from the source transducer [12]. Simultaneously, the directivity of the PA sounds is also almost at the same level as that of the primary sounds. Since the PA sounds are indirectly generated from the strong ultrasonic sounds by nonlinear interactions between waves at different frequencies, the net power efficiency for PA sound generation is extremely low. Hence, the power consumption of PA loudspeakers is high. Conventional PA loudspeaker systems on the market usually have a nominal power consumption of 65~100 W [4,8,9]. Although the power efficiency of the indirect generation process of PA sound through nonlinear acoustic interactions cannot be radically changed due to its inherent characteristics, there may be sufficient room in the power efficiency of PA source transducers, since the typical value is less than 15% for ultrasonic source transducers in air [11,13]. 

As discussed, Je et al. reported that a properly designed pMUT may exhibit up to 80% radiation power efficiency [10]. Since the typical power consumption of micro loudspeakers (excluding amplifier power consumption) adopted in mobile phones lies in the range of 0.5~1.2 W [14], this may be set as the criterion for the feasibility of applying PA loudspeakers to mobile devices. However, since a power amplifier circuit and a signal processing unit should be added to use the PA loudspeaker in a mobile device, the limit for the total power consumption of the PA loudspeaker system is set to be 5 W in this study, which includes the power consumption of the ultrasonic source transducer, the power amplifier circuits, and the signal processing unit. The pMUT source transducer, which possesses high radiation power efficiency, would lower the power consumption of the source transducer part sufficiently to considerably lower the power consumption of the power amplifier unit, and then the total power consumption of the PA loudspeaker system will be lower than the limit set in the above.

In the next three subsections, the pMUT array, power amplifier circuitry, and signal processing unit, which are designed and built for measuring total power consumption, are briefly described in the aspects of design. As can be seen, an appropriate combination of conventional units was adopted in the power amplifier and signal processing units while the PA source transducer was designed and fabricated as proposed in a previous study [5]. The fabrication techniques and processes for the pMUT array are described in Section 3.

### 2.1. Piezoelectric Micromachined Ultrasonic Transducer

#### 2.1.1. Design Requirements

The power consumption of a general micro loudspeaker for mobile phones is usually less than 1.5 W [15]. In a PA speaker, the linearity of the power amplifier is important because the non-linearity effect in the medium is exploited. Therefore, a class B type power amplifier with better linearity than other types of power amplifiers should be used. The theoretical efficiency of class B type power amplifiers is π/4 (≒78.5%). However, this efficiency decreases when DC and AC voltages are generated at the same time [15]. Assuming an efficiency for a class B amplifier of 50%, the power consumption of it would be around 3 W. Generally, the power consumption of a signal processing board is under 1 W. Therefore, the power consumption of the total system including the power amplifier and the signal processing board should be under 4 W for applying to a mobile device with a battery.

For this reason, we aim for a maximum power consumption of the pMUT array of under 1.5 W.

The PA source transducer radiates ultrasonic sounds of large amplitudes. Since the radiation efficiency is dependent on the ratio of the source diameter to the wavelength, the size of the unit transducer may be as small as the wavelength of the emitted ultrasonic sounds while the wavelength of 100 kHz sound is 3.43 mm.

A loudspeaker that can generate a primary wave (PW) with high intensity is required for generating a PA sound, which is very difficult because of the low acoustic radiation efficiency of the general loudspeaker. In air, it is not easy to generate the high-intensity sound in air required for such applications. The difficulty arises from the fact that the acoustic impedance of air (ρaircair = 415 Rayls) is generally much smaller than that of a solid-state transducer (*ρ_t_c_t_* = 34 MRayls for PZT4) [16]. This large impedance mismatch limits efficient transmission of energy from the transducer to air, and most of the energy is consumed by mechanical dissipation of the transducer. Therefore, improvement of the acoustic radiation efficiency of the transducer is essential for applying PA loudspeakers in mobile or portable devices with limited electric power.

A wide frequency bandwidth with a minimum of 10 kHz is required for playing instrumental and vocal sounds in the audible frequency range (20~20 kHz), which is similar to the range of conventional loudspeakers [17]. In 2013, Je et al. designed and fabricated a pMUT with high efficiency and a wide frequency bandwidth in air [10].

#### 2.1.2. pMUT Design

The pMUT can be modelled by a lumped parameter model (LPM), as shown in Figure 2. As described in Reference [10], the design purpose is to reduce the radiation quality factor to decrease the impedance mismatch between the pMUT and air. The mechano-acoustic efficiency ($\eta_{MA}$) of the pMUT can be expressed by the mechanical quality factor $Q_{m,}$ and the radiation-loaded quality factor $Q_{r}$.
(1)$$\eta_{MA}=\frac{R_{r}}{R_{m}+R_{r}}=\frac{Q_{r}}{Q_{m}+Q_{r}}$$
(2)$$Q_{m}=\frac{\sqrt{M_{eff}K_{eff}}}{R_{m}},\;Q_{r}=\frac{\sqrt{M_{eff}K_{eff}}}{R_{r}}$$
(3)$$Q=\frac{\sqrt{M_{eff}K_{eff}}}{R_{m}+R_{r}}=\frac{1}{\frac{1}{Q_{m}}+\frac{1}{Q_{r}}}$$
where $R_{0}$ is the electrical resistance, $R_{r}$ is the radiation resistance, and Q is the overall quality factor of the pMUT. Therefore, to achieve a large mechano-acoustic efficiency, the radiation quality factor should be much smaller than the unloaded mechanical quality factor, which can be achieved by reducing the size of the pMUT. To achieve a large frequency bandwidth, either the unloaded mechanical quality factor or the radiation quality factor should be small.

Figure 3 shows the calculated radiation quality factor of the pMUT as a function of the pMUT thickness when the resonance frequency of the pMUT was fixed at 100 kHz. The radiation quality factor decreases as the thickness decreases, and approaches a minimum of about 50 when the thickness is below 15 μm [10].

Figure 4 shows the calculated mechano-acoustic efficiency and overall quality factor of the pMUT as a function of the thickness. The overall quality factor decreases as the thickness of the pMUT decreases, and the minimum overall quality factor is approximately 50 when the thickness is less than 15 μm. Therefore, we may expect a maximum mechano-acoustic efficiency of 90% when the thickness of the pMUT is less than 15 μm and when resonance frequency of the pMUT is 100 kHz [10]. The determined parameters of the pMUT used in this paper are presented in Table 1.

The transducers are arranged in a triangular pattern at half-wavelength intervals, as shown in Figure 5. The half-wavelength arrangement avoids grating lobes in the beam pattern of the PW, and the mutual radiation effect between adjacent transducers can be ignored at the same time [10]. 

### 2.2. Signal Processing Board

#### 2.2.1. Signal Processing Board Requirements

To generate an audible acoustic signal using the PA phenomenon, both amplitude modulation (AM) and signal processing of low nonlinear distortion is required.

Since the proposed PA speaker system should be able to modulate an input acoustic signal into the ultrasound band in real time, a fast signal processing should be implemented, while maintaining a compact size for mobile use.

Modulation techniques to reduce nonlinear distortion have been reported since the first PA speaker-based AM was implemented in 1983 [1]. Among these modulations, we applied single sideband (SSB) modulation in our system. Since double sideband (DSB) modulation uses the lower and upper bands of the signal at the modulation frequency, the harmonic distortion caused by natural demodulation becomes large because of double bands. On the other hand, SSB, which uses only the upper part of the lower band for modulation, exhibits half the distortion of DSB modulation.

#### 2.2.2. Single Sideband Modulator Design for the PA Speaker

To implement SSB modulation, a Hilbert transformation is essential. Furthermore, fast computation of the Hilbert transformation is crucial for a real-time PA speaker system. 

To reduce errors in computation of the Hilbert transformation, increase computational speed, and implement algorithms into a signal chip, we adopted an infinite impulse response (IIR) all-pass filter and lattice filter structure for SSB modulation and implemented this into the signal processing board. 

The IIR filter structure for the Hilbert transformation was designed using an Eigenvalue problem [18]. The optimum IIR filter can be described by the Algorithm 1.

**Algorithm 1**
The optimum IIR filter algorithmChoose the initial extreme frequencies ω4(0)(k=0,⋯,N)For i=1,2,⋯
Compute P and Q, which have the following elements.Pij=W(ωi)sinαj(ωj),    (0≤j≤N)
Qij=(−1)i+lcosαj(ωj),    (0≤j≤N)Find the real maximum eigenvalue and a corresponding eigenvector.Search the extremal frequencies ωk(i)(k=0,⋯,N) for φ(ω), If ∑k=0N|ωk(i)−ωk(i−1)|<εBreak;end
end

The implemented IIR filter has lower distortion than a finite impulse response (FIR) filter with small filter taps. A comparison of IIR and FIR is shown in Figure 6. The proposed IIR filter has an almost flat all frequency band, while an FIR filter has a fluctuation and distortion band of 0–500 Hz. 

The final SSB modulator-based IIR filter is shown in Figure 7. The time delay block Z-(2N-1) is added for compensation of the time delay caused by the IIR filter, and r_o_ represents the reflection coefficient of the IIR filter.

The block diagram of the implanted digital signal processor (DSP) board is shown in Figure 8. The hardware (HW) is composed of a two-channel audio analog to digital converter, a fast DSP, and a one-channel digital-to-analog (DA) converter. The analog signal is converted to digital at a sampling rate of 36 Ksps and the digitized audio sample is received via a DSP McASP port (I2S format). The signal processing is composed of the serial processes of the stereo to mono signal conversion, lattice IIR filtering, 10× interpolation, and SSB modulation at a frequency of 90 kHz. Signal transmission is conducted by sending an interpolated 360 Ksps signal to the uPP port and DAC. The final stage of signal transmission is an analog low-pass filter with a cut-off frequency of 190 kHz to the DAC output.

### 2.3. Power Amplifier

#### 2.3.1. Power Amplifier Requirements

The power amplifier used in the PA speaker system should have a low total harmonic distortion (THD) and wide linear output bandwidth characteristics, and should maintain high power efficiency in the driving frequency band. For mobile applications, the dimensions must also be sufficiently small at approximately 7 cm. 

Furthermore, in this paper, the pMUTs used as the acoustic generator of the PA speaker show the best driving characteristics only when a DC bias voltage is applied, as shown in Figure 9b. However, many power amplifiers studied in the past are not suitable for driving pMUTs that require a DC bias because they operate on AC, as shown in Figure 9a. If a conventional AC power amplifier is directly applied to a pMUT, the power efficiency is lowered. Therefore, a large heat sink is required. This implies that a large-capacity power amplifier is required, which makes the overall system large in size and unsuitable for mobile applications. 

In addition, since the pMUT has a large capacitive reactance, a very large reactive power is generated when the pMUT is driven. Therefore, the power amplifier must supply an apparent power much larger than the power consumption of the transducer (active power). As a result, the power amplifier requires a larger capacity of elements in the implementation, and has lower power efficiency. Therefore, the power amplifier for driving the pMUT requires an impedance matching circuit capable of compensating the reactive power of the transducer. 

Therefore, in this paper, a new high-efficiency power amplifier optimized to drive the pMUT, and which satisfies the above-mentioned requirements, is implemented, and the design specifications of this power amplifier are provided in Table 2.

#### 2.3.2. Power Amplifier Design

##### Detailed Power Circuit of Proposed Amplifier

Figure 10 shows the overall block diagram of the proposed power amplifier, and Figure 11 shows the detailed power circuit diagram. The proposed power amplifier is composed of a power supply consisting of a boosted DC-DC converter that boosts the variable input voltage to a constant voltage, a class B amplifier that is a linear amplifier with excellent linearity to ensure sufficient total harmonic distortion (THD), and a matching circuit for improving the performance. The pMUT, as a load, is composed of two channels (CH1 and CH2). The power amplifiers used in PA speakers must prioritize low THD and the linearity of the output. Therefore, as shown in Table 3, a class-B amplifier with relatively low power efficiency but guaranteed linearity is applied to the PA speaker system. At the same time, by analyzing the electrical characteristics and power consumption of the pMUT, a matching circuit capable of canceling the reactive power generated by the transducer is added. Thus, the efficiency of the Class-B linear amplifier is maximized to minimize the power consumption of the proposed power amplifier.

To realize a high-efficiency power amplifier, the load characteristics of the pMUT must first be considered. Figure 12 shows the measured impedance magnitude, according to the DC bias voltage of pMUTs CH1 and CH2 used in this study. The impedance of the pMUTs is measured with driving frequency bands *f_1_* = 83 kHz and *f_2_* = 90 kHz. The impedance of pMUTs CH1 and CH2 can be confirmed to be constant without a large change when the DC bias voltage is 0~7 V. However, the impedance of the pMUTs increases slightly when the DC bias voltage exceeds 7 V. This impedance characteristic causes the output linearity of the transducer to be lowered. Therefore, it is appropriate to select the DC bias voltage of the power amplifier as the minimum voltage capable of driving the pMUT. Therefore, the DC bias voltage can be selected as shown in Figure 13.

Figure 13 shows the DC bias voltage and output waveform of the conventional power amplifier [5] and the proposed power amplifier. Class B type linear amplifiers generally have a drop-out voltage of 1.4 V and, thus, the proposed power amplifier is designed to have a margin of 2 V, because the drop-out voltage gradually increases with the output. Since the DC bias voltage is set to 7 V, the voltage V_EE_ supplied to the linear amplifier is 0 V, so that the power supply does not need the supply voltage V_EE_. The detailed circuit diagram of the proposed power amplifier considering the impedance characteristic of pMUT is shown in Figure 11.

##### Matching Circuit Design

Figure 14 shows the power amplifier and output waveforms with and without a matching circuit. As shown in Figure 14a, the pMUT has a large capacitive reactance, so that the output current i_o_ leads the output voltage V_o_ by a phase difference of θ. As a result, the reactive power Q_C_ is generated and the efficiency of the power amplifier becomes very low. Therefore, adding the matching circuit as shown in Figure 14b can compensate the reactive power generated by the transducer. Since the matching circuit consists of inductor L_m_ and DC blocking capacitor C_dc_ connected in series, as shown in Figure 14b, the impedance of the matching circuit is the sum of the reactance of inductor L_m_ and the reactance of capacitor C_dc_. If the capacitor C_dc_ is selected so that the reactance of the DC blocking capacitor C_dc_ becomes very small at the typical driving frequency of 86.5 kHz of the transducer, the impedance of the matching circuit at 86.5 kHz will be represented only by the reactance component of the inductor L_m_. At this time, the inductor L_m_, the transducer capacitor C, and the transducer resistor R form a parallel circuit structure. In this structure, if the inductor L_m_ is selected so that the magnitude of the reactive power Q_C_ generated by the capacitor C becomes equal to the magnitude of the reactive power Q_L_ generated by the inductor L_m_, the reactive power can be greatly reduced. As a result, the output voltage V_o_ and the output current I_o_ become equal in the phase, as shown in Figure 14b, and the apparent power at the output side is greatly reduced. Due to this characteristic, a power consumption amplifier with a matching circuit has higher power efficiency characteristics.

Figure 15 shows the measured capacitances of pMUTs CH1 and CH2 at a DC bias voltage of 7 V. Since the center frequencies of the transducers used in this study are 83 kHz and 90 kHz, the capacitances at the average frequency *f* = 86.5 kHz are C_CH1_ = 0.45 uF and C_CH2_ = 0.26 uF. Using the measured capacitances C_CH1_ and C_CH2_ of the pMUT, the inductance of the matching circuit is calculated as L_m_CH1_ = 6.5 uH and L_m_CH2_ = 11.2 uH.

Since the power amplifier in this study uses a DC bias voltage of 7 V, a DC blocking capacitor C_dc_ should be added to the matching circuit. However, to compensate the reactive power of the transducer, when the frequency is 86.5 kHz, the reactance size of the DC blocking capacitor C_dc_ should be much smaller than the reactance size of the inductor L_m_. The selected capacitor C_dc_ in this study is 22 uF. At the operating frequency of 86.5 kHz, the reactance magnitude of the inductor L_m_ is about 50 times larger than the reactance magnitude of the capacitor C_dc_.

Figure 16 shows the input power P_IN_ of the conventional power amplifier [5], and the input power P_IN_ of the proposed power amplifier, according to the output voltage amplitude when there is no matching circuit. For the case where there is no matching circuit, the proposed power amplifier has a lower input power when compared with the conventional power amplifier, and the input power of 0.8 W decreases when the output voltage V_O_ is 5 V.

Figure 17 shows the input power P_IN_ of the proposed power amplifier with and without a matching circuit. The input power of the proposed power amplifier with a matching circuit is reduced by 45% when compared with the case without a matching circuit, and, thus, the input power of 2.75 W is decreased when the output voltage *V_O_* is 5 V.

## 3. Fabrication and Packaging

### 3.1. pMUT Fabrication

The pMUT was bulk micromachined from a silicon substrate. A silicon-on-insulator (SOI) wafer with a 15-um-thick top Si layer, a 1-um-thick buried insulator, and a 500-um-thick handle layer was used to form a uniform membrane. The fabrication process is presented in Figure 18. First, thermal oxidation was performed to form an insulating layer between the PZT device and the silicon membrane, and a 2,500/330-Å-thick Pt/Ti layer was deposited using sputtering, as shown in Figure 18a. A 3-um-thick PZT layer was deposited using a sol-gel process, and a Pt/Ti layer was deposited on the PZT layer. Then, the electrodes and the PZT layer were etched as shown in Figure 18b. A photoresist (PR) was then patterned to form an air-gap, and gold lines and pads were used to form electrical contacts, as shown in Figure 18c. A perylene layer was coated on the front side to prevent an air-gap, as shown in Figure 18d, and the back side was then etched with aluminum window patterns using deep reactive ion etching (DIRE) to form the membrane, as shown in Figure 18e. Lastly, the pMUT was completed by releasing the perylene, as shown in Figure 18f.

The pMUT arrays were fabricated with array units that included 4 × 7 pMUT units for the control of performance uniformity and yield, as shown in Figure 19. The size of an array unit is 13 mm by 7 mm.

### 3.2. Packaged Compact Parametric Array Loudspeaker

A packaged compact PA loudspeaker system was assembled with a pMUT array, a digital signal processing (DSP) board, and a power amplifier. First, a total of 24 (4 × 6) array units of fabricated pMUTs were assembled on an aluminum jig of 9.7 cm × 8.6 cm and fixed with epoxy, as shown in Figure 20. The entire pMUT loudspeaker was composed of 672 units, with a radiated area of 42 mm × 52 mm. 

There are printed circuit boards (PCBs) for three electrodes (two signal and one ground) on the edge of the pMUT array on the aluminum jig, which are connected with a card edge-type PCB on the bottom of the aluminum jig, as shown in Figure 20.

A DSP board and power amplifier board are fabricated (card edge-type) as shown in Figure 21. The size of the fabricated DSP board and power amplifier are 8 cm × 9 cm and 6 cm × 7 cm, and they are connected together by a PCB with a card edge-type connector, as shown in Figure 22. A packaged compact PA loudspeaker system (PALS) was built in an aluminum case, as shown in Figure 23. A battery (5V, 2A) was installed to supply power for the circuits under the aluminum case.

## 4. Results and Discussion

### 4.1. Admittance Measurement

The admittance curve of the fabricated pMUT array was measured in air using an impedance analyzer (4294A, Agilent), as shown in Figure 24. The black solid line indicates the conductance, and the gray dotted line is the susceptance. The resonance frequencies were around 83 and 90 kHz with 20% lower than the 100 and 110 kHz designed resonance frequencies. This is because the diameter was bigger than the designed value due to the footing effect in the micro-electromechanical systems (MEMS) fabrication process [19].

### 4.2. Acoustic Experimental Setup

An acoustic measurement experiment was performed in 3 m × 3 m × 2 m semi-anechoic chamber, as shown in Figure 25. The pMUT array is installed on a rotation stage. The input signal is generated by a function generator (33522A, Agilent), which is modulated by the AM method in an internal DSP board in PALS. This signal was generated on *f_1_* and *f_2_* units individually using the out-of-phase method in an internal two-channel power amplifier. The acoustic waves radiating from the pMUT array were measured using a calibrated high frequency pressure-field microphone (1/8-in. microphone, type 4138, B&K). The measured data was visualized using an oscilloscope (TDS 2024B, Tektronix) and a dynamic signal analyzer (SR785, SRS).

### 4.3. Driving Signal

Two *f_1_* and *f_2_* PWs are required to generate a difference frequency wave (DFW). The frequency of the DFW is determined by the difference between the two PW frequencies (*f_d_* = *f_2_*−*f_1_*). The input signals are applied on each channel of the pMUT array, as shown in Figure 5.

Input signals are generated below.
(4)$$\begin{array}{l} V_{1}=V_{DC}+V_{AC}\left\{\sin\left(2\pi f_{1}t\right)+\sin\left(2\pi f_{2}t\right)\right\}\\ V_{2}=V_{DC}-V_{AC}\left\{\sin\left(2\pi f_{1}t\right)+\sin\left(2\pi f_{2}t\right)\right\}\\ \left(\;-:\;\mathrm{out}-\mathrm{of}-\mathrm{phase}\right) \end{array}$$

*V_1_* and *V_2_* are the driving signals to generate the *f_1_* and *f_2_* pMUT units, and V_DC_ and V_AC_ are generated by 7 V and 5 Vp, where the (−) sign of *V_2_* indicates the out-of-phase *V_1_* driving signal.

### 4.4. Acoustic Characteristics Measurement

#### 4.4.1. Propagation Curve

To evaluate the acoustic characteristics of the PW, the SPL of the PW was measured. The measurements were performed using a sinusoidal wave in out-of-phase driving configurations. Figure 26 shows the experimental SPL results of the PW measured in air at 0.45 m, which is similar to the actual driving environment. The sound pressure in the 82–90 kHz range for the PW was about 125 dB, and its deviation was within ±3 dB.

To evaluate the acoustic characteristics of the DFW generated by the PA, the SPL of the DFW was measured at a distance of 0.45 m. Figure 27 shows the frequency response of the SPL on the DFW. At this time, the sound pressure of the PA sound in the audible frequency range was about 62 dB. The deviation was within ±3 dB and the linear frequency bandwidth was confirmed at 12.5 kHz in the audio frequency band.

The propagation curve measures the SPL of the PW and DFW as a function of the distance from the device. The propagation curves of the PW at a carrier frequency, *f_c_* of 90 kHz, were measured. In the case of the propagation curves for the DFW, the transducer was operated in the out-of-phase configuration (Equation (1)), where the PW signal *f*_1_ was operated at 89, 85, and 80 kHz, while *f*_2_ was maintained at 90 kHz. This generated DFWs at a frequency, *f_d_*, of 1, 5, and 10 kHz, which were subsequently measured. Figure 28 shows the measured PW and DFW propagation curves of the packaged compact PALS.

#### 4.4.2. Beam Pattern

Figure 29 shows the measured radiation beam pattern of PW at 83 kHz and 90 kHz at 0.45 m away from the pMUT array. It was confirmed that the half power bandwidth (HPBW) forms a beam at 6° with 83 kHz DFW and at 6° with 90 kHz of PW.

Figure 30 shows the radiation beam pattern of DFW at 5 and 10 kHz at 0.45 m away from the pMUT array. Driving signals in the out-of-phase configuration were applied similarly to the propagation curve measurements. It was confirmed that the HPBW forms a beam at 7.3° at 5 kHz DFW and 7.1° at 10 kHz DFW.

### 4.5. Power Consumption Measurement

Figure 31 shows the power consumption measurement system. The power consumption of the pMUT at *f_c_* was calculated by the generated voltage and measured conductance curve. The voltage and current of the DSP board, the power amplifier, and the packaged compact PALS were measured using the DC power supply shown in Figure 6, and their power consumptions were calculated as follows.
(5)P=VI
where *P* is the input electric power, *V* is the input voltage, and *I* is the current.

Table 4 shows the power consumption of the components of the PALS.

The power consumption of the pMUT array was measured to be up to 1.29 W when driven at the *f_c_*. The power consumption of the DSP board was measured to be 0.5 W. The power consumption of the packaged compact PALS was measured to be 6.65 W without a matching circuit. After the matching circuit described in the previous chapter was added, the power consumption decreased by 3.4 W, which achieved a total power consumption of only 3.25 W.

## 5. Conclusions

In this paper, we developed a packaged compact PALS. A pMUT array was fabricated using a MEMS process, and an SSB modulator as a signal processing board and a Class B power amplifier were applied on the PALS. Although the resonance frequencies of the pMUT radiator were shifted by about 20 kHz in the negative direction from the designed values due to problems in the micromachining fabrication process, the test results of the fabricated units confirm the possibility of using PA loudspeakers for mobile systems. Specifically, the manufactured PALS achieved 6.65 W without a matching circuit, and 3.25 W with a power matching circuit. The main problems in the fabrication process for the pMUT radiator are believed to be caused by footing effects in the deep reactive ion etching (DRIE) process, and mitigation of footing effects will form the basis of future work. 

The Class B type power amplifier used in the PALS has good linearity but somewhat lower power efficiency. If a Class D type power amplifier with higher efficiency is designed and applied to the PALS, it is expected that the power consumption of the PALS would be lower than 2 W, which is about two times larger than the loudspeaker is in the speaker phone mode of the conventional smart phone. 

In this case, we have confirmed the possibility of using PALS for mobile applications. The packaged PA speaker system developed in this study is applicable not only in laptops and televisions, but also in limited power portable device audio systems. In order to use the pMUT PALS in the smartphone, the signal processing units and the power amplification unit should be integrated into one package of the pMUT array in the form of Integrated Circuits.

The SPL of the audible sounds need to be enhanced. It can be done by a design improvement of the pMUT structure and the uniformity control in the fabrication process of pMUT, especially the DRIE process to release the membrane. The improvement in the uniformity of the pMUTs in the array will also provide progress in beamforming and reduction of the side or grating lobes. Additional improvement in beamforming will be achieved by adopting the independent control on the signals to drive each pMUTs.

## Figures and Tables

**Figure 1 sensors-19-04449-f001:**
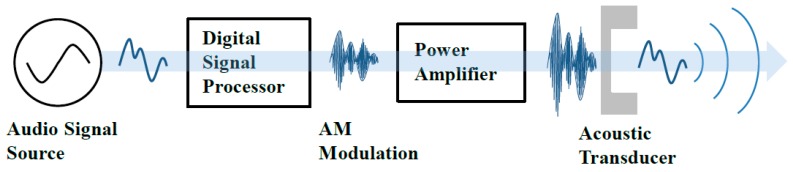
Parametric array (PA) loudspeaker system.

**Figure 2 sensors-19-04449-f002:**
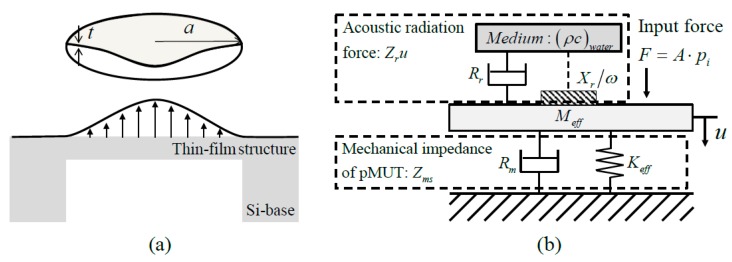
(**a**) Schematic and (**b**) lumped parameter model (LPM) of piezoelectric micro-machined ultrasound transducer (pMUT).

**Figure 3 sensors-19-04449-f003:**
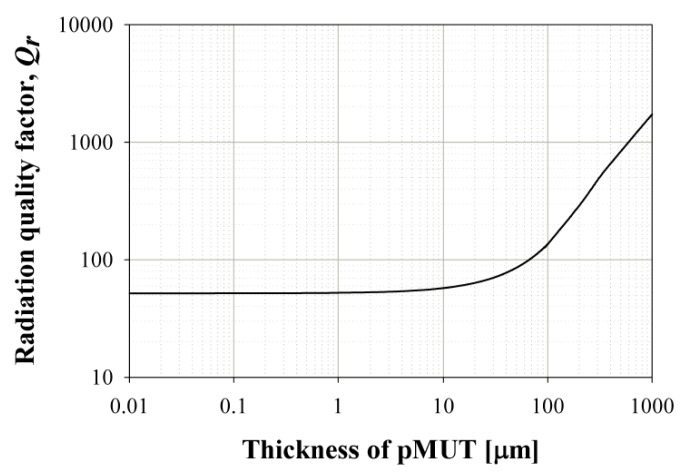
The radiation quality factor depends on thickness of pMUT.

**Figure 4 sensors-19-04449-f004:**
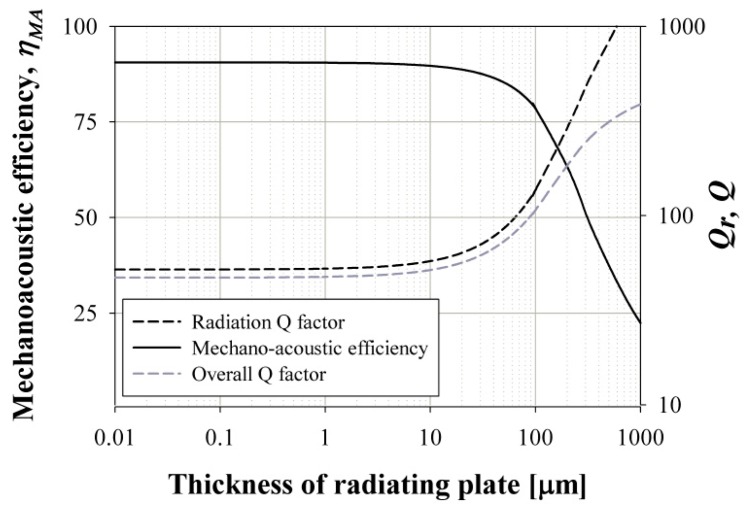
Mechano-acoustic efficiency and the overall quality factor depend on thickness of pMUT.

**Figure 5 sensors-19-04449-f005:**
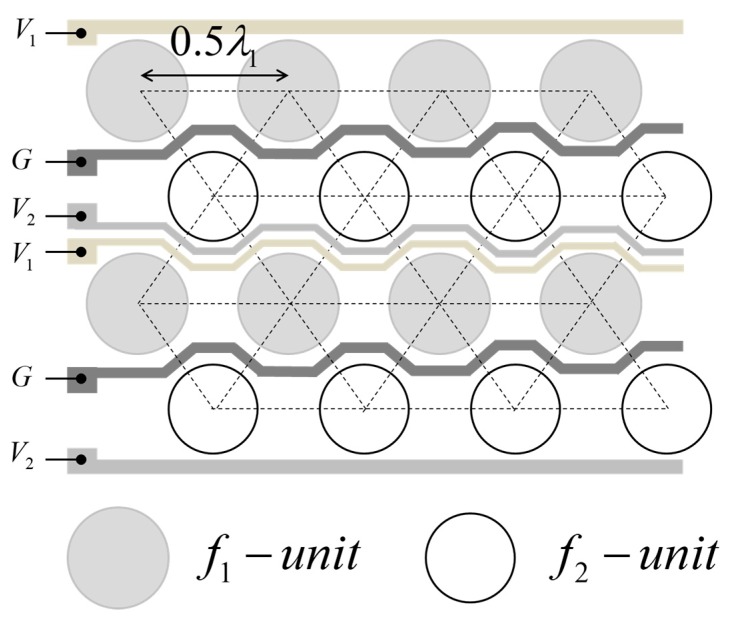
Piezoelectric micro-machined ultrasound transducer (pMUT) array geometry and signal electrodes.

**Figure 6 sensors-19-04449-f006:**
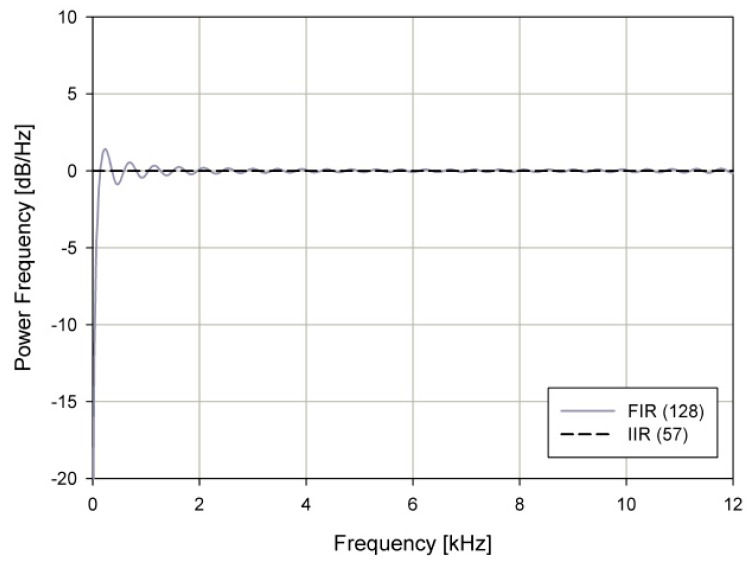
Frequency response of an infinite impulse response (IIR) and finite impulse response (FIR) filter.

**Figure 7 sensors-19-04449-f007:**
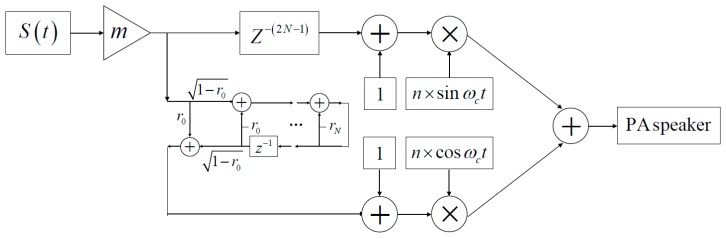
The proposed single sideband (SSB) modulator.

**Figure 8 sensors-19-04449-f008:**
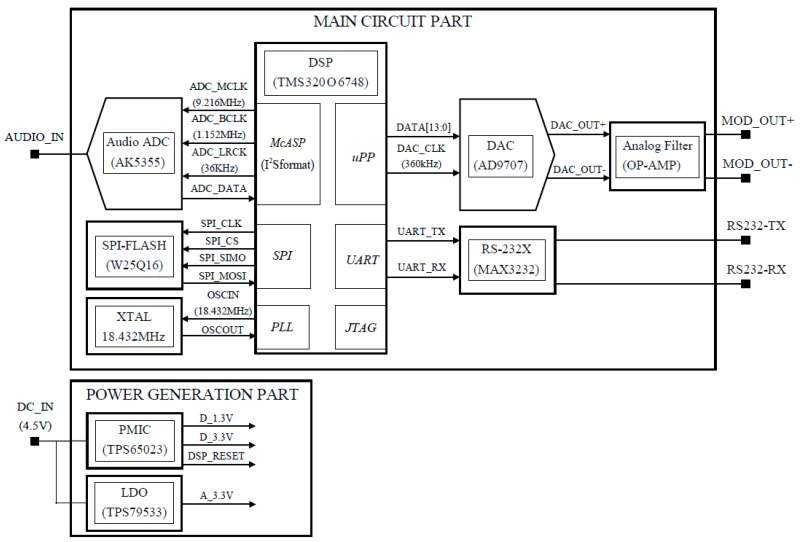
Block diagram of the digital signal processor (DSP) board.

**Figure 9 sensors-19-04449-f009:**
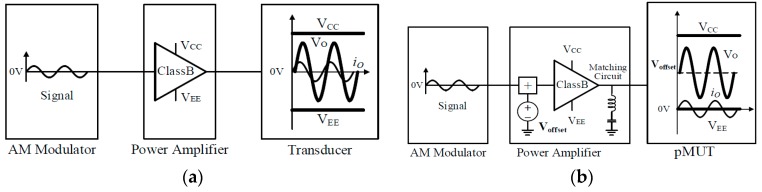
Basic characteristics of the power amplifier for (**a**) typical transducers, and (**b**) piezoelectric micromachined ultrasonic transducers (pMUTs).

**Figure 10 sensors-19-04449-f010:**
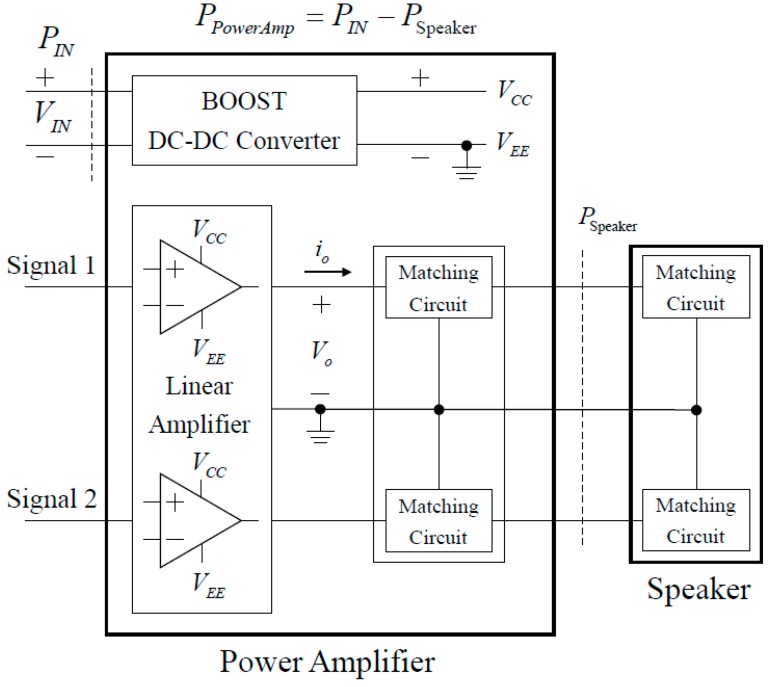
Overall block diagram of the proposed power amplifier.

**Figure 11 sensors-19-04449-f011:**
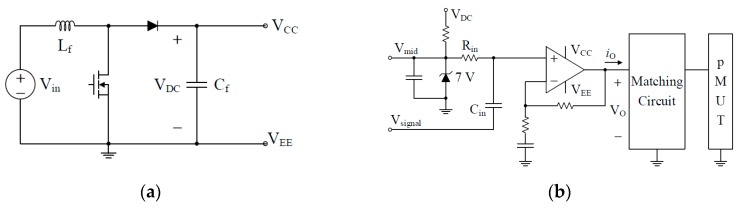
Detailed power circuit of the proposed power amplifier. (**a**) Boost direct-direct (DC-DC) converter, and (**b**) linear amplifier and matching circuit.

**Figure 12 sensors-19-04449-f012:**
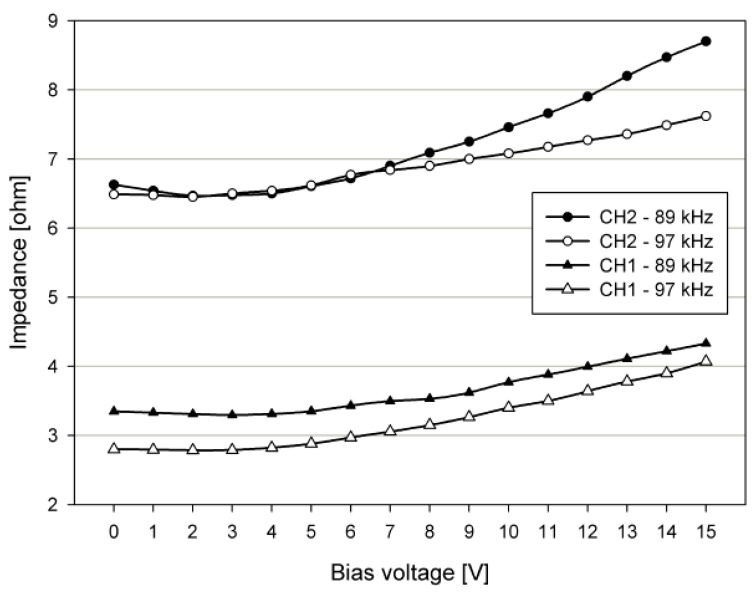
Measured impedance of piezoelectric micro-machined ultrasound transducers (pMUTs) CH1 and CH2.

**Figure 13 sensors-19-04449-f013:**
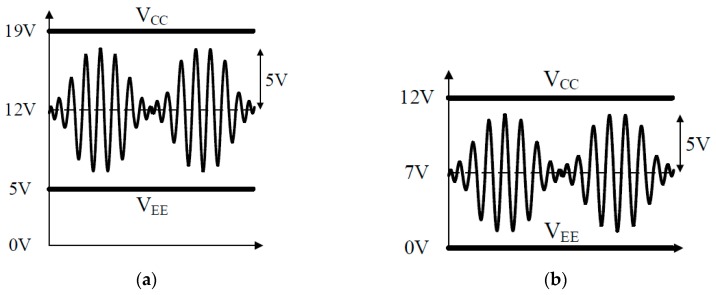
Detailed power circuit of the proposed power amplifier. (**a**) boost DC-DC converter, (**b**) linear amplifier, and matching circuit.

**Figure 14 sensors-19-04449-f014:**
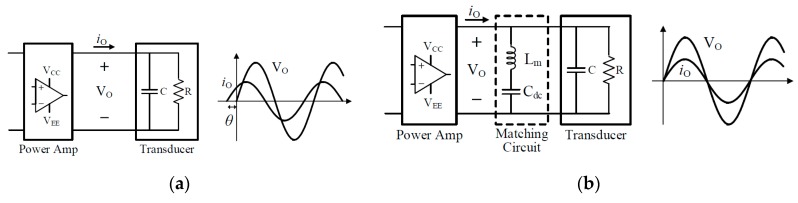
The output waveform of the power amplifier (**a**) without the matching circuit, and (**b**) with the matching circuit.

**Figure 15 sensors-19-04449-f015:**
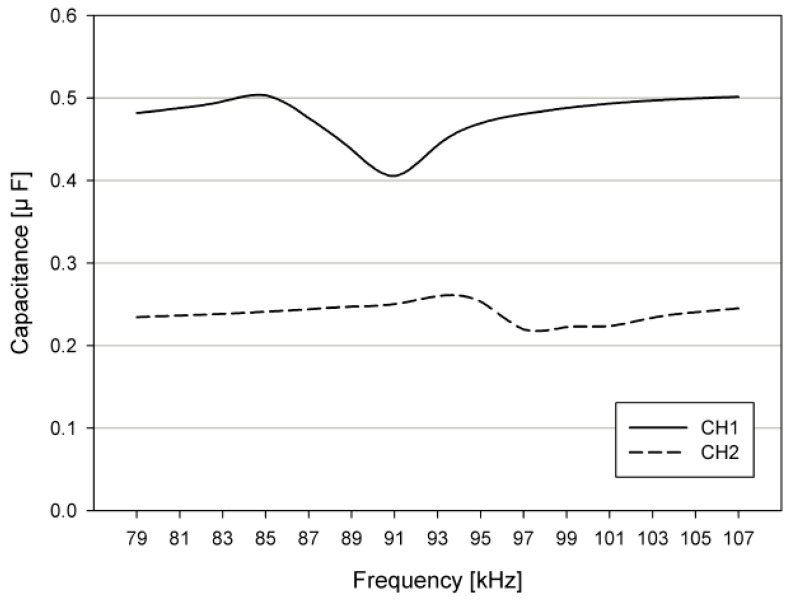
Measured capacitances of pMUTs CH1 and CH2, according to the driving frequency.

**Figure 16 sensors-19-04449-f016:**
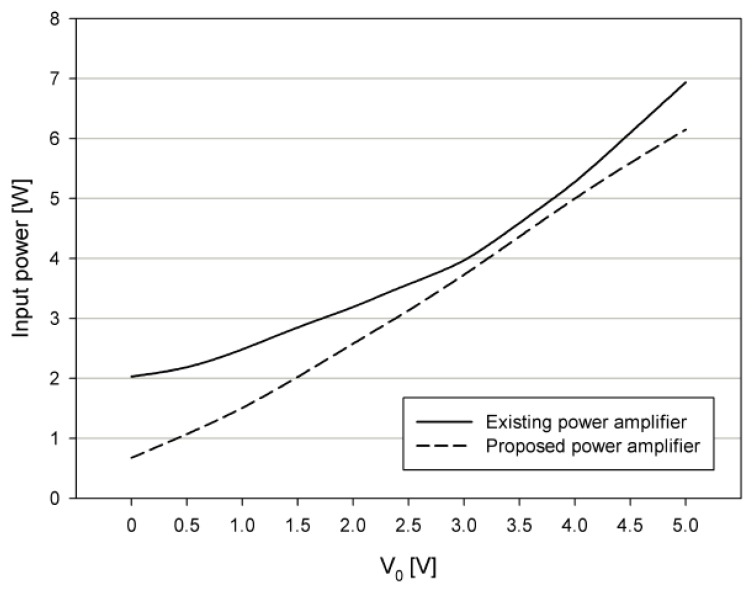
Input power of the existing power amplifier [5] and proposed power amplifier without a matching circuit.

**Figure 17 sensors-19-04449-f017:**
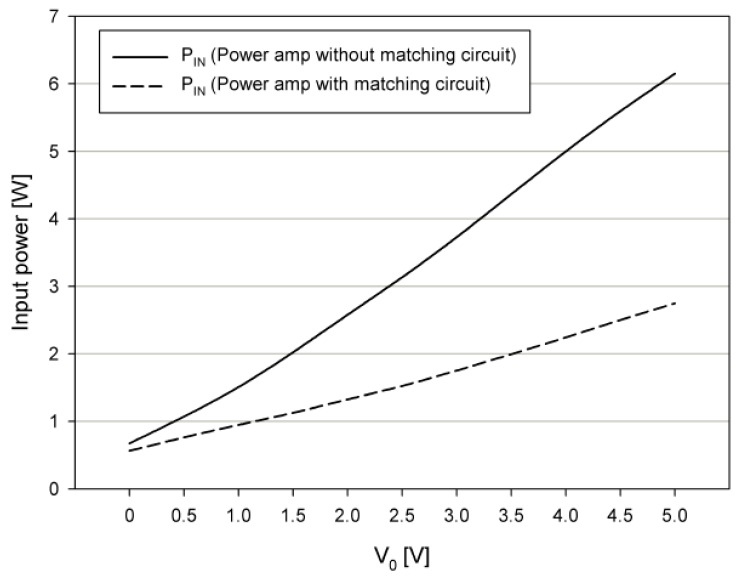
Input power of the proposed power amplifier according to the presence or absence of the matching circuit.

**Figure 18 sensors-19-04449-f018:**
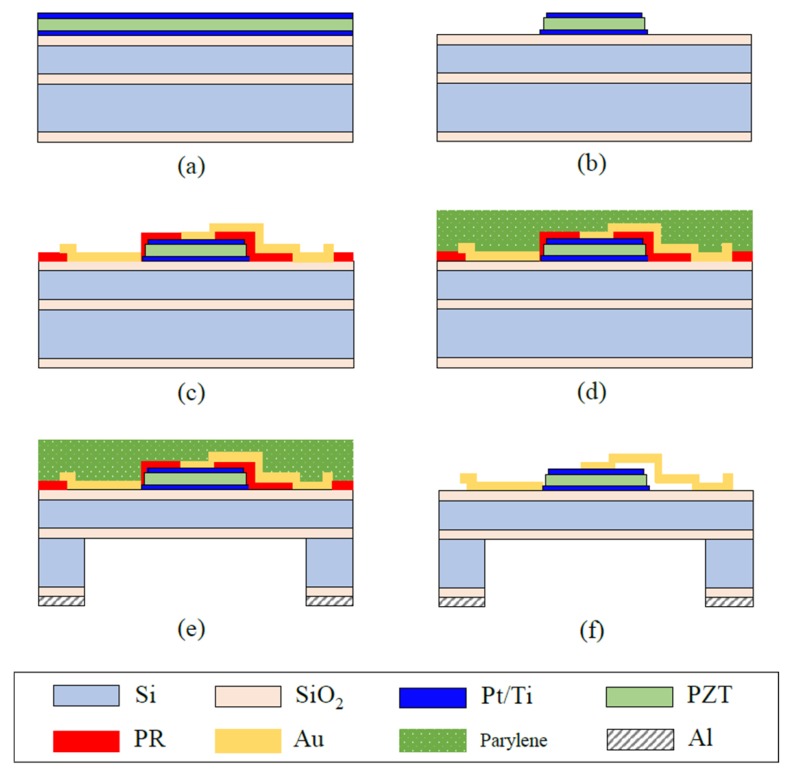
Fabrication process of a piezoelectric micro-machined ultrasound transducer. (**a**) Bottom electrode/PZT/top electrode deposition, (**b**) Patterning of bottom/PZT/top electrode, (**c**) Gold electrode deposition and patterning, (**d**) Perylene passivation on the front side, (**e**) The DRIE process, (**f**) Releasing the perylene coat.

**Figure 19 sensors-19-04449-f019:**
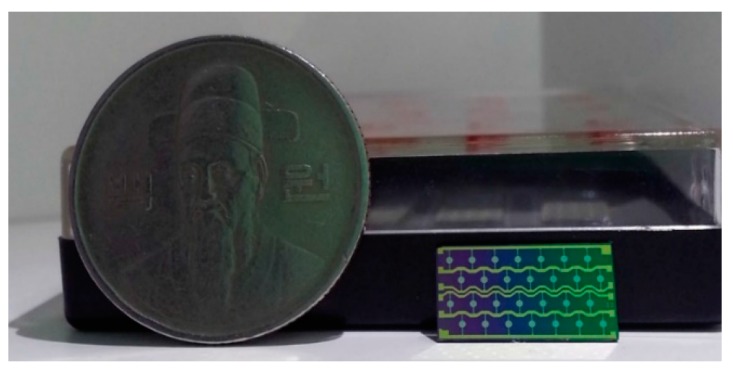
Piezoelectric micro-machined ultrasound transducer (pMUT) array unit (4 × 7).

**Figure 20 sensors-19-04449-f020:**
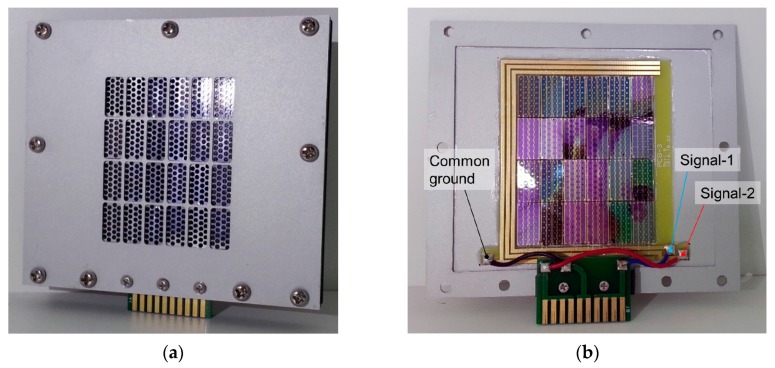
The output waveform of the power amplifier (**a**) without the matching circuit, and (**b**) with the matching circuit.

**Figure 21 sensors-19-04449-f021:**
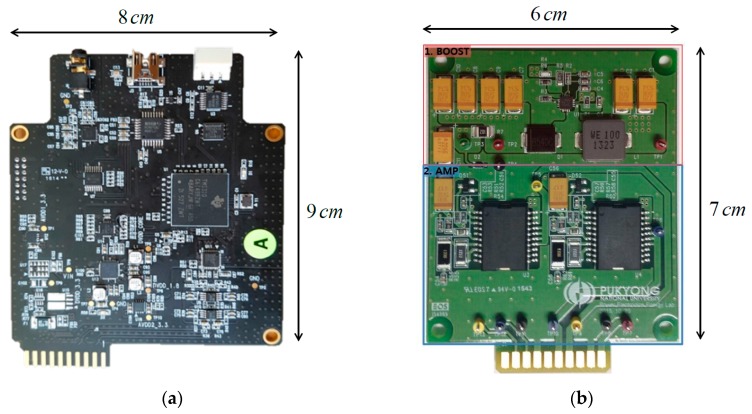
Fabricated (**a**) digital signal processing (DSP) board, and (**b**) power amplifier board.

**Figure 22 sensors-19-04449-f022:**
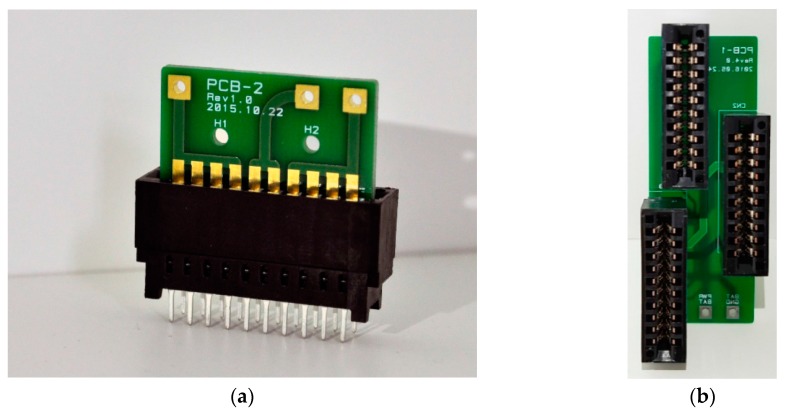
(**a**) Card edge connector, (**b**) PCB with card edge connector.

**Figure 23 sensors-19-04449-f023:**
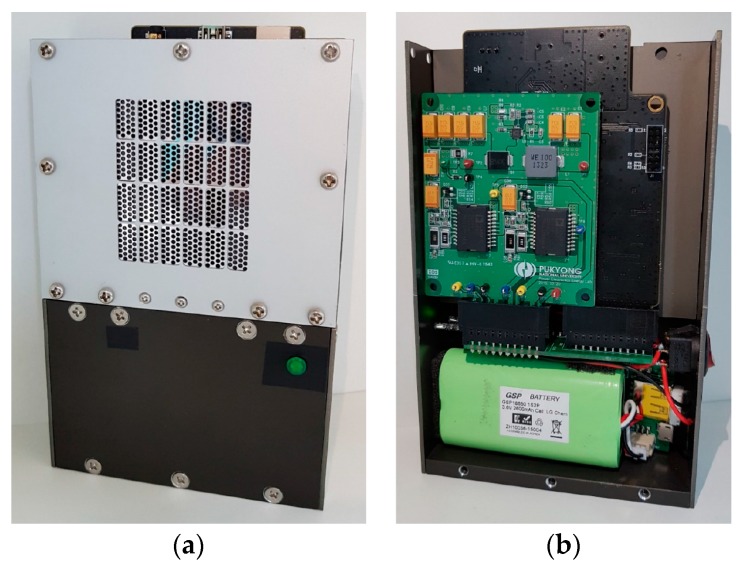
(**a**) Packaged parametric array loudspeaker system, and (**b**) the inside appearance.

**Figure 24 sensors-19-04449-f024:**
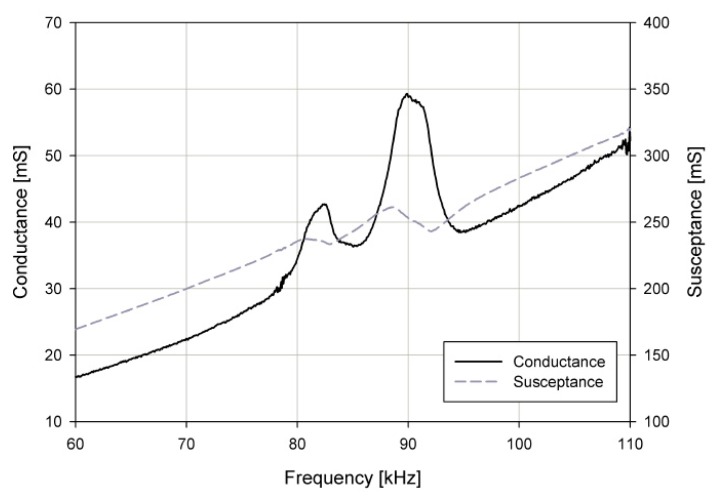
Admittance measurement graph: conductance (solid line) and susceptance (dotted line).

**Figure 25 sensors-19-04449-f025:**
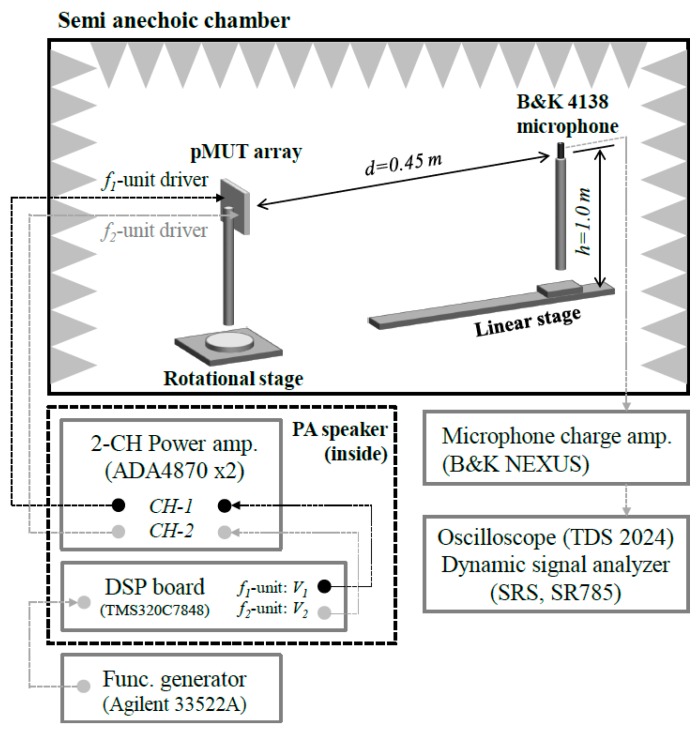
Acoustic measurement experimental environment.

**Figure 26 sensors-19-04449-f026:**
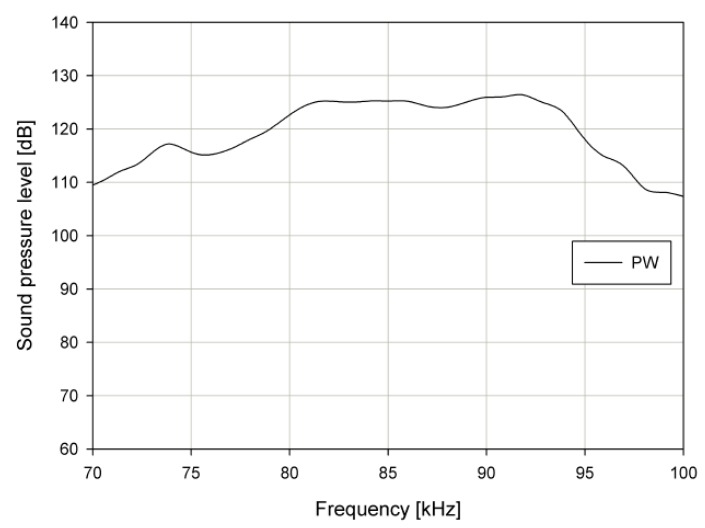
Sound pressure level (SPL) of primary wave (PW).

**Figure 27 sensors-19-04449-f027:**
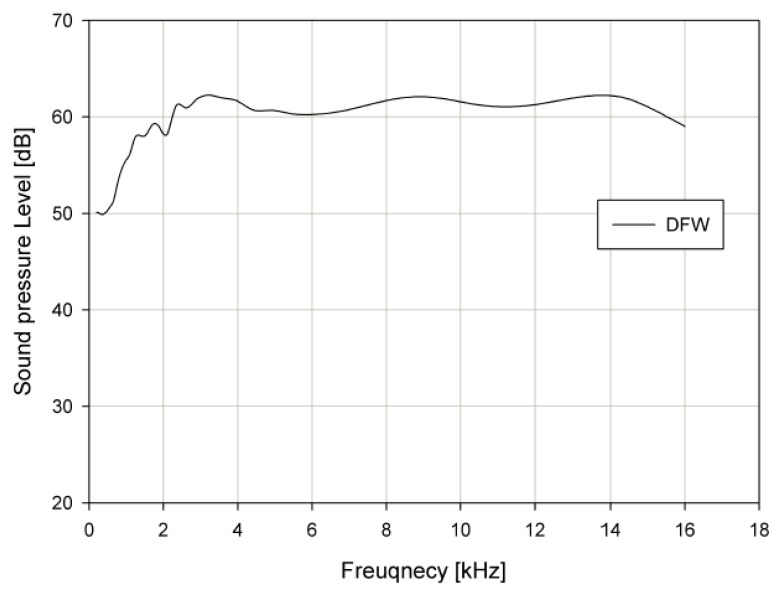
The SPL of difference frequency wave (DFW).

**Figure 28 sensors-19-04449-f028:**
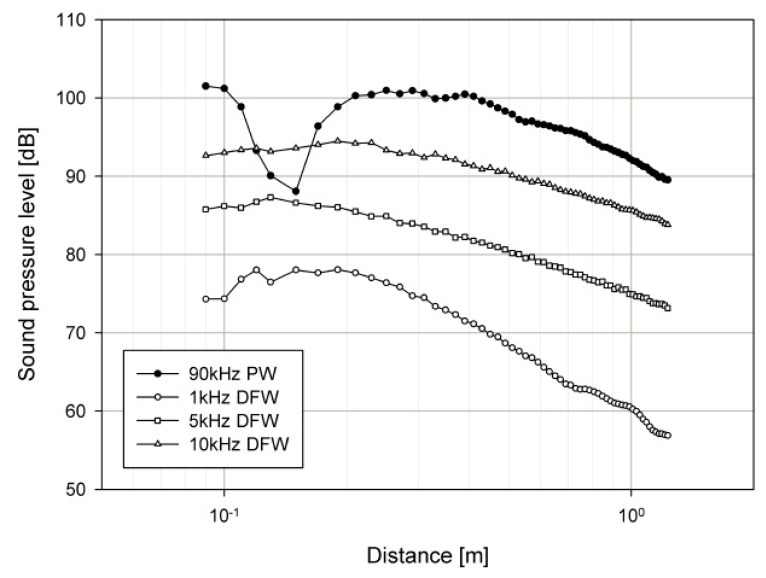
Measured propagation curves for the PW and the DFWs.

**Figure 29 sensors-19-04449-f029:**
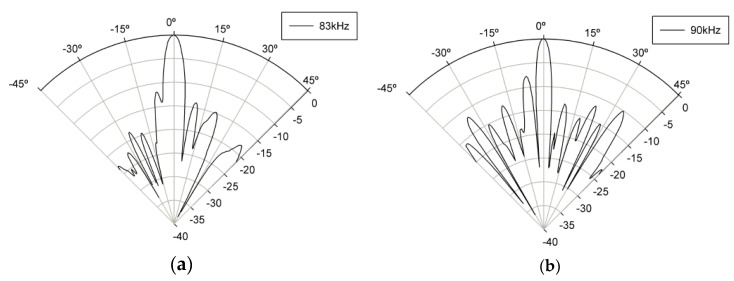
Beam pattern of the primary wave (PW) at (**a**) 83 kHz and (**b**) 90 kHz.

**Figure 30 sensors-19-04449-f030:**
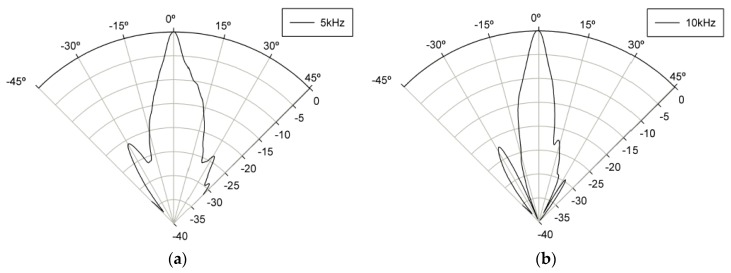
Beam pattern of DFW at (**a**) 5 kHz and (**b**) 10 kHz.

**Figure 31 sensors-19-04449-f031:**
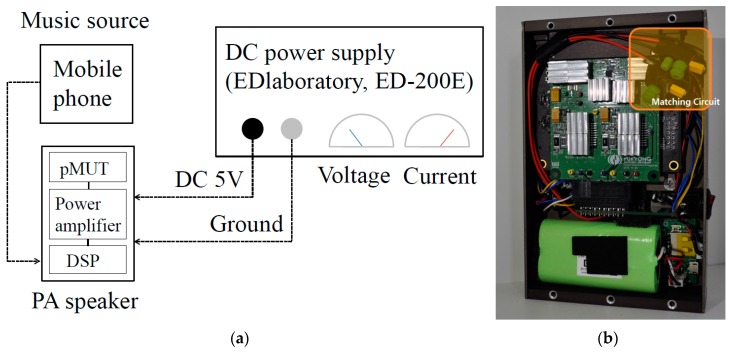
(**a**) Power consumption measurement, and (**b**) packaged PA loudspeaker system with a matching circuit.

**Table 1 sensors-19-04449-t001:** Geometry of piezoelectric micro-machined ultrasound transducer (pMUT) units.

	Resonance Frequency	Radius, *a*	Thickness, *t*
*f_1_*-pMUT unit	100 kHz	725 μm	15 μm
*f_2_*-pMUT unit	110 kHz	687 μm	

**Table 2 sensors-19-04449-t002:** Power amplifier specification.

Parameters	Thickness, t
Input power rating	10 (W)
Input voltage	5–12 (V)
Input signal frequency *f*_1_, *f*_2_	83, 90 (kHz)
Boost output voltage V_cc_	14 (V)
Reference voltage of amplifier	7 (V)
Boost output voltage V_EE_	0 (V)
Maximum peak output voltage range of the amplifier	+2 to +12 (V)

**Table 3 sensors-19-04449-t003:** Comparison of power amplifier characteristics.

Linear Amplifier (class A, AB, B)	Non-linear Amplifier (class C, D, E, etc.)
High output linearity	Low output linearity
Low power efficiency (~78.5%)	High power efficiency (~95%)

**Table 4 sensors-19-04449-t004:** Power consumption of the components of the PA loudspeaker system.

	pMUT (at *f_c_*)	DSP Board	Power Amplifier	PALS
w/o matching circuit	1.29 W	0.5 W	6.15 W	6.65 W
w/ matching circuit			2.75 W	3.25 W

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
