# Peer review of "A Critical Step to Using a Parametric Array Loudspeaker in Mobile Devices"

_sensors, 2019, doi:10.3390/s19204449_

Round 1

Reviewer 1 Report

This paper presents PMUT array as loundspeaker with better coupling and reduced power consumption. Overall it is well organized but below comments need to be addressed.

-- array design, figure 5: this half lambda pitch is based on 100-110 kHz. why like that? human audio frequency is <20 kHz.

--fabrication process, figure 18. why metal trace designed to be air bridge in f)? Doesn't have reliability concern? any advantage?

--4x7 array design? any data to support 4x7 array desgin? and it looks only 4 channels. are they shorted or seperate for super array?

--directivity: beam pattern presented in this paper is 2D or 3D? if 2D it is on which direction of 4x6 super array?

Reviewer 2 Report

Authors suggested the possibility of PALS based on pMUT for mobile applications. The overall research is well-designed and the designed experiment was performed properly. But, several limitations of the PALS for the use in the mobile phone were found in the results. Authors should discuss how to overcome the limitations for mobile applications. 

For example, the power consumption of the PALS was too high to apply it with a current form for mobile applications. To apply the proposed system to mobile devices, a lot of works may need to be done in the future. In this study, authors only claimed the possibility of PALS based on pMUT for mobile applications. Therefore, authors should discuss how it is possible to apply it for mobile applications more clearly and which future works have to be done for it. What was the target directivity of the pMUT array? I could find that many side and grating lobes in the primary beam patterns. Also, asymmetric beam patterns were found. How to do beamforming? Authors may want to do such beamforming more carefully to achieve better beampatterns. I am sure that it can be improved.  
